# Relationship between Oral Health Knowledge and Maternal Oral Health with Obstetric Risk and Breastfeeding

**DOI:** 10.3390/ijerph19137797

**Published:** 2022-06-25

**Authors:** Silvia Serrano-Sánchez, Jaime González-González, Beatriz Rodríguez-Martín, Vanesa Muñoz-Rodríguez, Sonia de las Heras-Corrochano, Juan José Criado-Alvarez

**Affiliations:** 1Castilla-La Mancha Health Service, 45600 Talavera de la Reina, Spain; sserranos@sescam.jccm.es (S.S.-S.); vamuro@sescam.jccm.es (V.M.-R.); sdlas@sescam.jccm.es (S.d.l.H.-C.); 2Department of Medical Sciences, Faculty of Health Sciences, University of Castilla-La Mancha, 45600 Talavera de la Reina, Spain; jaime.gonzalez@uclm.es (J.G.-G.); juanjose.criado@uclm.es (J.J.C.-A.); 3Department of Nursing, Physiotherapy and Occupational Therapy, Faculty of Health Sciences, University of Castilla-La Mancha, Avd/Real Fábrica de Sedas s/n, 45660 Talavera de la Reina, Spain; 4Department of Health, Institute of Health Sciences, 45600 Talavera de la Reina, Spain

**Keywords:** periodontal disease, preterm birth, dental caries, breast feeding, low birth weight

## Abstract

The relationship between maternal gingival health status and low birth weight or preterm delivery is controversial. The aim of this study was to analyze the association between maternal oral knowledge and the level of oral health during pregnancy with the risk of obstetric complications and breastfeeding. A descriptive cross-sectional study was conducted after an oral health educational intervention in a consecutive sample of 97 pregnant women. Data collection consisted of a validated questionnaire, oral examination, the Caries Index (CAOD) and the Simplified Oral Hygiene Index (IHOS). The participants had a mean age of 32.5 ± 5.19 years and a predominantly university education (57.1%). The level of knowledge regarding oral health was fair (12.5 ± 3.56 correct answers). Older pregnant women (33.0 ± 4.80 years) practiced breastfeeding and had a higher number of correct answers to the questionnaire. Adequate IHOS was associated with higher birth-weight newborns (3333 ± 0.3), whereas poor oral hygiene control was associated with lower birth-weight newborns (2960 ± 0.1) (*p* < 0.05). A lower level of academic education was associated with worse oral hygiene (*p* < 0.05). In addition, the greater the number of children, the higher the CAOD. Finally, among non-smoking women, the weight of infants was 437 mg higher. Maternal oral hygiene and the week of delivery were associated with newborn weight (*p* < 0.05) in a multiple linear regression model. Smoking was also related to low birth weight (*p* < 0.05). Educational interventions in pregnancy are necessary to decrease the incidence of obstetric adverse effects and improve the oral health of mothers and their children.

## 1. Introduction

There is a relationship between oral cavity pathologies and certain cardiovascular and respiratory diseases, diabetes, pregnancy disorders, rheumatoid arthritis [1,2] and Alzheimer’s disease [3,4]. In addition, periodontal disease increases the risk of tumor development due to the high and continuous presence of inflammatory mediators derived from bacterial aggression in periodontitis [5]. Likewise, periodontitis has been associated with an increased risk of complications, ICU admissions and death in people with COVID-19 [6,7,8].

Pregnancy causes physiological and hormonal changes in women, affecting the endocrine, pulmonary, cardiovascular, gastrointestinal and hematological systems, among others [9]. Although many studies have analyzed the relationship between periodontal diseases and pregnancy, there is some controversy about the relationship between the gingival health status of the mother and the risk of low birth weight or preterm delivery [10,11]. In addition, hormonal changes during pregnancy cause changes to the oral cavity [12] such as erosion of the dental enamel [13], physiological xerostomia [14], gingivitis gravidarum [15] and epulis gravidarum [16]. In addition, there is an increased inflammatory response of periodontal tissues to local irritants by estrogens and progesterone that modifies vascular permeability, favoring edema [9]. According to the 2005 Oral Health Survey in Spain [17], periodontitis affects 25% of pregnant women. This data coincides with previous reviews that place the prevalence between 5 and 20% [18]. Periodontal disease is currently considered an “emerging risk factor” for obstetric complications. These mainly include low birth weight (<2500 g), preterm birth (<37 weeks), miscarriage, stillbirth and pre-eclampsia (commonly defined as maternal hypertension and proteinuria after the 20th week of gestation) [19]. However, there is no consensus on whether this association is consistent, because previous studies show biases in patient characteristics and some divergence in the criteria used in the periodontal study [20].

In recent years, we have witnessed an increase in preterm births and low-birth-weight babies [21]. It is estimated that between 15% and 20% of children born worldwide have low birth weight (20 million newborns each year), which is a public health problem and is associated with short- and long-term consequences. In Spain, one of the risk factors present in almost half of premature births (about 14,000 cases) could be maternal periodontal disease [22,23].

Several systematic reviews [24,25] associate periodontitis in pregnancy with the risk of preterm delivery or the birth of low-birth-weight babies. Periodontal infections constitute a portal for the spread of bacteria that can reach the fetoplacental unit and jeopardize its stability [26,27,28]. This type of disease constantly releases Gram-negative anaerobic bacteria, bacterial products, endotoxins, lipopolysaccharides and inflammatory mediators (PGE-2 and TNF alpha) into the bloodstream that, once disseminated by the transplacental hematogenous route, could lead to intrauterine infection–inflammation [29]. The increased production of inflammatory cytokines and c-reactive protein may contribute to uterine contraction, leading to miscarriage, premature delivery or low birth weight [30].

Most clinical studies indicate a positive correlation between periodontal disease and possible complications in pregnancy due to microbiological and immunological factors [31,32,33,34,35,36,37]. According to the 2005 Oral Health Survey in Spain [17], 25% of pregnant women present periodontitis; these data coincide with other reviews that suggest a prevalence of between 5 and 20% [18].

In 2004, the American Academy of Periodontology recommended periodontal evaluations for all pregnant women or those who are planning a pregnancy. We know that appropriate preventive and therapeutic care should be provided at this stage [12], as evidence shows that dental prophylaxis during pregnancy is safe, in addition to reducing bacteria and inflammation [38].

The rates of abandonment of breastfeeding in our setting are still high, without reaching the objectives of the international recommendations in this regard. Sociodemographic variables influence success.

The main aim of the study was to analyze whether there is an association between the level of knowledge on oral health, maternal oral health status and breastfeeding and the possible influence on gestational age and neonatal weight.

## 2. Materials and Methods

A descriptive cross-sectional study was conducted in the Talavera de la Reina Health Area (Toledo, Spain), at an oral health care office of the Talavera-Centro Health Center (Castilla-La Mancha Health Service, SESCAM), between October 2017 and October 2019. A consecutive sample of pregnant women who attended the birth preparation sessions given by the midwife of the Health Center was used for data collection. These women were referred to the Oral Health Unit of the center, where they were informed of the study. In our area, there are no racial or ethnic differences that might bias or influence the results of the study.

The sample size was calculated using the Epidat Version 4.1 program. Since the prevalence of the clinical characteristics and profile of the patients and the probability of knowledge about oral health in pregnant women were unknown, the maximum indeterminacy was assumed, which is a prevalence of 50%. With an assigned population of 120 pregnant women, and using a precision of 7.5%, with a confidence level of 95% (*p* < 0.05), a minimum sample size of 71 patients was obtained. A non-response or non-cooperation rate of 10% was assumed; therefore, the sample size would be increased by seven patients. The estimated total for the minimum sample was 78 patients. Ultimately, the sample size was 97 patients. Consecutive sampling methods were applied for the women arriving at the Oral Health Unit. 

### 2.1. Study Description

Data collection included sociodemographic data, medical history and an odontogram. We used the questionnaire on oral health for pregnant women, which is a validated questionnaire [39,40,41,42,43] consisting of 22 dichotomous and multiple-choice questions on knowledge about oral health prevention (eight items), oral diseases (six items), pregnancy and oral health (four items) and dental development (four items). The questionnaire provided one point for each correct answer, and the results were grouped into the following intervals: Poor (0–6 points), Fair (7–13 points), Good (14–20 points). The questionnaire also collected general information on age, trimester of pregnancy, educational level and previous visits to the dentist. The group of questions on oral health prevention integrated content on preventive measures such as tooth brushing, fluoride application, nutrition and hygiene. To evaluate knowledge on oral diseases, several questions were included on gingivitis, periodontal disease, caries and bacterial plaque. Regarding gestation and dental care, participants were asked about the use of anesthesia, taking dental X-rays during pregnancy and pediatric dental care. Regarding dental development and growth, concepts on temporary dentition were included, such as chronology of eruption, types of dentition and tooth formation.

After responding to the survey, the participants attended a talk/presentation entitled “Oral health and pregnancy. First oral care of the child”. This talk had a duration of 40 min and included questions and answers at the end.

The Caries Index (CAOD) and the Simplified Oral Hygiene Index (IHOS) were used to evaluate the oral health status of pregnant women. The CAOD is a cumulative epidemiological index that describes the caries experience of an individual or a population from eruption to the day of examination of the permanent dentition. It is determined by adding carious teeth (C), absent teeth (A) due to caries and filled teeth (O) and dividing this figure by the number of individuals studied. The IHOS is proposed by the WHO to evaluate the state of oral hygiene, assessing the presence and quantity of plaque and dental calculus, classified from 0 to 3. It determines and evaluates the degree of oral hygiene in the population, both quantitatively and qualitatively [44]. This index consists of two elements: a simplified debris index (DI-S) that describes the extent of soft deposits and is one of the two components of the simplified oral hygiene index (IHOS). The DI-S is obtained per person by totaling the debris score per tooth surface and dividing the result by the number of surfaces examined. In addition, the simplified calculus index (CI-S) is calculated. The average individual or group DI-S and CI-S are combined to obtain the IHOS. A mouth mirror and dental exploration probe are used for the examination. 

No disclosing agents were used. The six tooth surfaces examined were the vestibular surfaces of the right upper first molar, the right upper central incisor, the left upper first molar and the left lower central incisor. Further, the lingual surfaces of the left lower first molar and the right lower first molar were examined. There was only one observer in this study (S.S.S.), who is a qualified dentist with fifteen years of professional experience.

Newborns were subsequently examined by the midwife at the health unit fifteen days after birth. During this visit, a general examination of the newborn and the mother was performed, including the newborn’s birth weight as reflected in the hospital medical record, and the mother was given a survey about breastfeeding.

### 2.2. Statistical Analysis

For the descriptive analysis, descriptive parameters were used according to the scale of the variable; qualitative variables were calculated using means of frequencies and percentages, and quantitative variables were resolved using central position statistics (mean) and standard deviation. The distribution analysis was analyzed using the Kolmogorov–Smirnov and Levene tests for the study of the normal distribution of the variable with the Lilliefors significance level, in addition to verifying the symmetry of the curve or observing the frequency histograms with the normal curve. For the inferential analysis, in the case of the analysis of independent variables, ANOVA was used to study the relationship between a normal continuous variable and a nominal variable or the study of n independent groups with the correction of Bonferroni; if the outcome variable was a dichotomous variable, the student’s *t* test was used. To compare nominal and dichotomous variables, the Chi-square test was used. Multiple linear regression models were constructed following the backward procedure, using infant weight as the dependent variable and the rest of the variables studied as independent variables. Multiple linear regression models were constructed following the backward procedure, using infant weight as the dependent variable and the rest of the variables studied as independent variables. The backward procedure was performed with the selection of a significance level to stay in the model (*p* < 0.05), fitting the model with all possible predictors and comparing these after removing the predictor.

The existence of autocorrelations has been verified by studying the Durbin–Watson (DW) statistic, which ranges from 0 to 4 and takes the value of 2 when the residuals are independent. Values less than 2 indicate a positive autocorrelation, and those greater than 2 indicate a negative autocorrelation. Independence between the residuals can be assumed when 1.5 ≤ DW ≤ 2.5. Collinearity was analyzed with the tolerance and IVF statistics. For no multicollinearity, the tolerance must be high; a tolerance less than 0.10 diagnoses serious collinearity problems. Moreover, IVF is a reciprocal indicator of tolerance. The lower this value, multicollinearity must be reduced. It is estimated that an IVF value greater than 10 is diagnostic of serious collinearity problems. A confidence level of 5% was established. The SPSS statistical program for Windows (Statistical Package Social Sciences version 29.0, IBM Corp., Armonk, NY, USA) and Epidat 4.1 for Windows were used for data analysis.

### 2.3. Ethical Aspects

This study complies with the Declaration of Helsinki and with all ethical principles and data protection legislation. The study was approved by the Clinical Research Ethics Committee (CEIC) of the Gerencia de Atención Integrada de Talavera de la Reina (Registration 03/2017). All participants signed the informed consent form. 

## 3. Results

Ninety-seven pregnant women participated in the study. Table 1 summarizes the main characteristics of the participants.

Breastfeeding was reported by 84.8% of the women, and 5.2% of the women smoked during pregnancy. Most of the women visited the dentist at least once (97.9%), and 11.5% were receiving dental treatment at the time. The main problems for which they visited the dentist on their last visit were prophylaxis (35.8%), general consultation (20%) and endodontic treatment (16.8%). The main reasons for which they did not visit the dentist were: “I don’t have any pain in my teeth” (35.4%), “I am pregnant” (23.1%) and “Financial reasons” (13.8%).

The mean CAOD score was 7.1 ± 3.58 points (Median: 8.0; Range: 0–18), whereas the IHOS index had a mean 1.0 ± 0.72 points (Median: 0.9; Range: 0.1–4.1). The level of oral health knowledge of the pregnant women was fair (mean 12.5 ± 3.56 correct). The mean percentage of correct and incorrect answers for each dimension was calculated as a function of the total number of questions in each dimension (Figure 1). The dimensions or areas of knowledge with the best results, with values of 69% and 67% of correct answers, respectively, were those related to preventive measures and oral pathologies. In contrast, the dimensions dealing with pregnancy and oral health, as well as knowledge of dental development, scored the lowest, with a failure rate of 55% and 58%, respectively. 

In relation to knowledge about prevention, the item related to knowledge of the harmful habit of putting the child to sleep with a bottle in their mouth (93.8%), followed by the relationship between digital/pacifier sucking habits and possible future problems affecting dentition, were the most accurate. The item with the most incorrect responses was related to the recommended age of the child’s first dental visit (66%). Regarding knowledge about oral pathologies, the item with the most correct answers was related to the popular saying “for every pregnancy a tooth is lost”; thus, 96.9% of the participants considered this statement to be incorrect. The item with the most errors (78.7%) was considering caries to be a contagious disease. In the area of knowledge on the relationship between pregnancy and oral pathologies, the item with the most correct answers was the one referring to the relationship between medications taken during pregnancy and their possible future dental influence on the baby. In the area of knowledge on dental development, the item with the most correct results (68.4%) was related to the moment of eruption of the primary teeth, and the item with the worst results (88.4%) questioned the moment during pregnancy when the baby’s teeth begin to form. 

Table 2 shows the results of different variables according to whether breastfeeding was maintained or not. Maternal age was higher in those women who practiced breastfeeding, with 33.0 ± 4.80 years, versus those who did not, with 29.6 ± 6.42 (*p* < 0.05). We also found a higher percentage of correct answers in the questionnaire in women who breastfed, with 12.8 ± 3.40 correct answers, compared to those who did not, with 10.7 ± 4.02 questions (*p* < 0.05). For the remaining variables, no statistically significant differences were found (*p* > 0.05).

In addition, we found a statistically significant relationship between the level of maternal oral hygiene and infant weight. Thus, an adequate IHOS was related to heavier newborns (3333 ± 0.3), whereas poor oral hygiene control was related to lower birth weight (2960 ± 0.1). The results show a statistically significant relationship *(p* < 0.05) between the IHOS questionnaire scores and the mother’s level of education; thus, the lower the level of education, the higher the IHOS, signifying worse oral hygiene (Table 3).

In the normal analysis of variance (ANOVA) of the variables studied according to the level of studies, the statistically significant variable (*p* < 0.05) was the IHOS; thus, a lower level of studies was related to a higher index (1.6), compared to university studies with an index of 0.77 ± 0.509. The other variable that was statistically significant *(p* < 0.05) was the mother’s age (Table 4).

In the correlation between the CAOD and the different quantitative variables, we only found a statistically significant association (*p* < 0.05) between the CAOD and the number of children, although the Pearson correlation coefficient was not very relevant (0.236). The Pearson correlation coefficient between IHOS and infant weight (R = −0.33), maternal age (R = −0.386) and weeks of pregnancy (R = −0.219) had a statistically significant association (*p* < 0.05), with no relationship with the remaining variables (Figure 2).

In the multiple linear regression model (Table 5 and Figure 2), the variables that were statistically significant (*p* < 0.05) and related to infant weight were IHOS (Coefficient: −0.169), indicating that there was an inverse relationship between IHOS and birth weight (*p* < 0.05). Meanwhile, the week of delivery had a direct relationship with birth weight (Coefficient: 0.138) (*p* < 0.05). In Figure 2, the independent variable IHOS and birth weight are shown.

An estimated 5.2% of the sample smoked at the time of the study. Table 6 shows the results of different variables according to smoking habit. Statistically significant differences (*p* < 0.05) were only observed in the case of newborn weight, with a difference of 437 milligrams in favor of women who did not smoke.

## 4. Discussion

The general oral health knowledge of pregnant women was fair, a finding that is in line with previous studies using the same scale [39,40,41,45]. As in Correa’s study [46], knowledge about prevention is the dimension with the best score. Furthermore, our results coincide with Vozza et al.’s study, confirming the lack of knowledge of the vertical transmission of cariogenic bacteria through saliva and the fact that the pregnant women in the sample did not consider caries to be a contagious disease [47]. As in other studies [39,48], the timing of tooth formation in pregnancy is the item that was answered incorrectly by the most pregnant women (88.4%). However, the item with the most correct answers (96.9%) was the one related to suffering dental loss due to pregnancy. This is a very positive finding, since it constitutes a very deep-rooted popular false belief, as previously reported by other authors [49,50,51].

As previously mentioned, there is a lack of knowledge regarding the safety of the use of dental anesthesia during pregnancy, dental X-rays and the most appropriate trimester to receive dental care [39,42,49,50,51]. 

In our population, 97.9% of women had visited the dentist at least once. The reasons for consultation were conservative or preventive treatments. This is a positive finding compared to the other studies reviewed, in which the frequency of visits to the dentist was lower, and the main reason for consultation was pain [45,52,53]. 

The CAOD index score is high and could be improved in our group. Even so, it is lower than the study by García-Martín et al. that highlighted the negative impact of the presence of caries during pregnancy [54]. As in a previous study, the CAOD index appears to be linked to cultural and socioeconomic level. Thus, the higher the economic level, the lower the incidence and prevalence of dental caries [55]. Of the participating pregnant women, 5.4% had a smoking habit during pregnancy. Our results agree with those of Castaldi et al., as we failed to find an association with low birth weight or preterm delivery. Rather, it is the combination of these two circumstances (poor periodontal health and smoking) that is associated [56]. 

The oral hygiene index observed in our group was acceptable (Mean: 1 ± 0.72), revealing a statistically significant relationship with infant birth weight, maternal age and weeks of pregnancy. This finding is supported by the results obtained by other authors relating maternal periodontal health status with infant weight and the week of delivery [31,32,33,34,35,36,37,56,57,58]. The IHOS index was used because it is recognized by the WHO for this type of study. Furthermore, it is noninvasive and easy to explore and record, since it only requires one observer. It would be interesting to use other periodontal indexes in future studies. 

In view of our results, we can affirm that the percentage of women who breastfeed is high (84.5%). Further, in agreement with other authors, we can affirm that sociodemographic variables influence the success of breastfeeding [59,60,61]. A limitation of our study is that the moment at which breastfeeding was documented was too early to assess established breastfeeding, and no distinction was made between exclusive breastfeeding and mixed breastfeeding. 

The novelty of our study is the consideration of relating breastfeeding with variables related to the mother’s oral health. Thus, we found that women who breastfeed had the highest rate of correct answers in the questionnaire used, “Questionnaire for pregnant women. Oral health.” The strengths of our study include the fact that we explored the presence of caries and measured the oral hygiene of pregnant women using tools that are recognized by the WHO and are widely used in other epidemiological studies because of their simplicity, clarity and ease of application [62,63,64].

In line with other authors, it is necessary to expand the preventive expectations of oral health within current health policies. Thus, the most effective method of intervention is the application of health education programs, especially the development of specific protocols in oral health and pregnancy, with the participation of dentistry, obstetrics and primary care professionals [54,65]. Prevention based on a solid knowledge of the oral health of parents through maternal education in birth and pregnancy preparation programs translates into better oral health indices in their children [39,66]. 

## 5. Conclusions

The level of knowledge on oral health among pregnant women attending birth preparation sessions is only fair. Some basic misconceptions persist, which should be addressed. Maternal oral hygiene and smoking are associated with birth weight and week of delivery, although the cause–effect relationship has not yet been demonstrated. The higher the level of maternal oral hygiene, the higher the birth weight of the baby. Oral check-ups in pregnant women are essential. A preventive oral health program for pregnant women should be implemented, under the coordination of both midwives and oral health units. Consequently, it is necessary to strengthen the link between professionals.

Breastfeeding was established in most of our sample at the time of the postpartum check-up (15 days). However, a long-term follow-up several months later would be necessary to confirm whether breastfeeding continued. 

In the field of oral health during pregnancy and related knowledge, there is a need for educational interventions in order to mitigate risk factors and thus reduce the incidence of adverse obstetric effects and help to improve the oral health of mothers and their future children. Further epidemiological studies are needed, as is the allocation of more professional and economic resources for this purpose, considering the existing biases due to characteristics of the participants (socioeconomic, cultural, biological, environmental variables...), to obtain further evidence and work with unified protocols, criteria and common methodologies that allow the results to be verified. This would enable health education based on risk factors to be offered at an early stage. 

Oral health education during pregnancy is still an unresolved subject, and there is a need for unified guidelines and protocols with a multidisciplinary approach.

## Figures and Tables

**Figure 1 ijerph-19-07797-f001:**
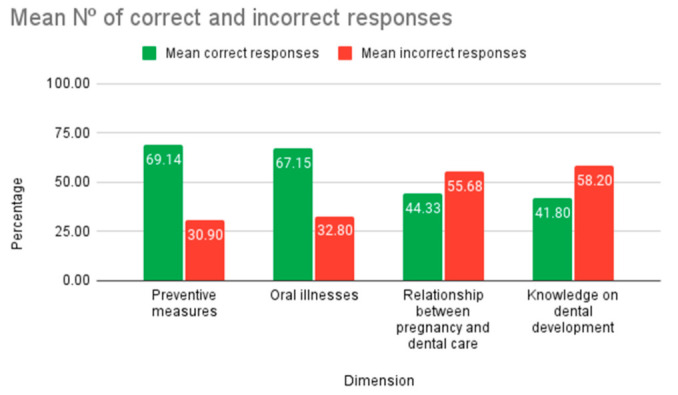
Bar chart of percentages of correct and incorrect answers by dimensions.

**Figure 2 ijerph-19-07797-f002:**
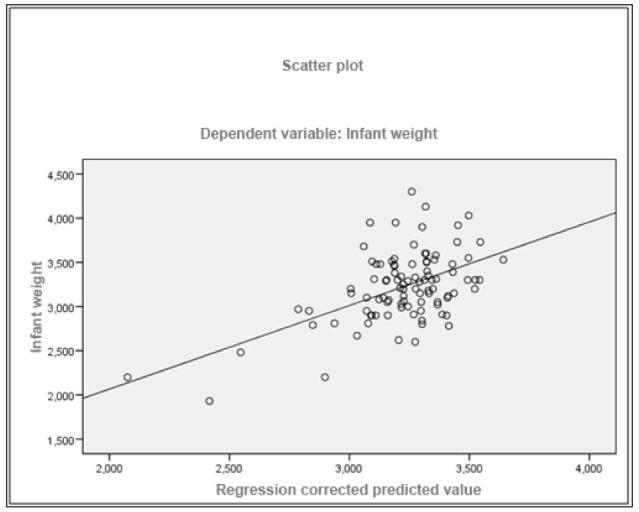
Scatter plot considering birth weight as the dependent variable.

**Table 1 ijerph-19-07797-t001:** Main characteristics of the participants.

Age in Years	32.5 ± 5.19 Years (Median: 33; Range: 18–45)
**Educational level**	52 (57.1%) university studies 38 (41.8%) high school education 1 (1.1%) primary education.
**Number of children**	1.4 ± 0.63 (Median: 1; Range: 1–3)
**Week of delivery**	39.1 ± 1.36 week (Median: 39; Range: 33–41)
**Newborn weight**	3221.1 ± 0.39 milligrams (Median: 3200; Range: 1930–4300)

**Table 2 ijerph-19-07797-t002:** Results of the different variables according to breastfeeding habits.

	Breastfeeding	F Frequency	Mean ± SD	Statistical Significance
**Caries Index (CAOD) score**	**Yes**	82	7.18 ± 3.615	*p* > 0.05
	**No**	15	6.93 ± 3.515	
**Simplified Oral Hygiene Index (IHOS) score**	**Yes**	82	1.04 ± 0.739	*p* > 0.05
	**No**	15	1.17 ± 0.637	
**Week of delivery**	**Yes**	82	39.1 ± 1.44	*p* > 0.05
	**No**	15	39.3 ± 0.81	
**Infant’s weight (milligrams)**	**Yes**	82	3221 ± 0.41	*p* > 0.05
	**No**	15	3220 ± 0.34	
**Number of Children**	**Yes**	82	1.5 ± 0.63	*p* > 0.05
	**No**	15	1.4 ± 0.63	
**Mother’s age in years**	**Yes**	82	33.0 ± 4.80	***p* < 0.05**
	**No**	15	29.6 ± 6.42	
**Weeks of Pregnancy**	**Yes**	82	31.0 ± 5.35	*p* > 0.05
	**No**	15	30.8 ± 6.38	
**Correct answers “Questionnaire for pregnant women. Oral health”**	**Yes**	82	12.8 ± 3.40	***p* < 0.05**
	**No**	15	10.7 ± 4.02	
**Incorrect answers “Questionnaire for pregnant women. Oral health** **”**	**Yes**	82	8.5 ± 3.33	*p* > 0.05
	**No**	15	10.2 ± 3.26	

SD: Standard Deviation.

**Table 3 ijerph-19-07797-t003:** Influence of the variables studied on the Simplified Oral Hygiene Index (IHOS).

	IHOS Control	Frequency	Mean ± SD	Statistical Significance
**Caries index (CAOD)**	**Suitable**	63	6.95 ± 3.674	*p* > 0.05
	**Acceptable**	32	7.50 ± 3.538	
	**Deficient**	2	7.50 ± 0.707	
	**Total**	97	7.14 ± 3.582	
**Week of delivery**	**Suitable**	63	39.2 ± 1.17	*p* > 0.05
	**Acceptable**	32	39.0 ± 1.72	
	**Deficient**	2	39.5 ± 0.70	
	**Total**	97	39.1 ± 1.36	
**Infant’s weight (milligrams)**	**Suitable**	63	3333 ± 0.3	***p* < 0.05**
	**Acceptable**	32	3017 ± 0.3	
	**Deficient**	2	2960 ± 0.1	
	**Total**	97	3221 ± 0.3	
**Number of children**	**Suitable**	63	1.5 ± 0.6	*p* > 0.05
	**Acceptable**	32	1.5 ± 0.6	
	**Deficient**	2	1.0 ± 0.0	
	**Total**	97	1.4 ± 0.6	
**Age**	**Suitable**	63	34.0 ± 4.28	***p* < 0.05**
	**Acceptable**	32	29.5 ± 5.76	
	**Deficient**	2	33.0 ± 0.0	
	**Total**	97	32.5 ± 5.19	
**Weeks of pregnancy**	**Suitable**	63	31.4 ± 5.24	*p* > 0.05
	**Acceptable**	32	30.3 ± 5.73	
	**Deficient**	2	28.0 ± 11.31	
	**Total**	97	31.0 ± 5.49	

SD: Standard Deviation. IHOS: Oral Hygiene Index.

**Table 4 ijerph-19-07797-t004:** Normal analysis of variance (ANOVA) of the variables studied according to educational level.

	Level of Education	Frequency	Mean ± SD	Statistical Significance
**Caries index (CAOD)**	**Primary**	1	8.00	*p* > 0.05
	**Secondary**	38	7.55 ± 3.562	
	**University**	52	6.73 ± 3.609	
	**Total**	91	7.09 ± 3.574	
**Simplified Oral Hygiene Index (IHOS)**	**Primary**	1	1.60	***p* < 0.05**
	**Secondary**	38	1.39 ± 0.834	
	**University**	52	0.77 ± 0.509	
	**Total**	91	1.04 ± 0.729	
**Week of delivery**	**Primary**	1	40.0	*p* > 0.05
	**Secondary**	38	39.2 ± 1.38	
	**University**	52	39.1 ± 1.22	
	**Total**	91	39.2 ± 1.28	
**Infant weight (mg)**	**Primary**	1	3290	*p* > 0.05
	**Secondary**	38	3205 ± 0.3	
	**University**	52	3253 ± 0.3	
	**Total**	91	3233 ± 0.3	
**Number of children**	**Primary**	1	1.0	*p* > 0.05
	**Secondary**	38	1.5 ± 0.64	
	**University**	52	1.4 ± 0.64	
	**Total**	91	1.4 ± 0.63	
**Mother’s age**	**Primary**	1	18.0	***p* < 0.05**
	**Secondary**	38	31.1 ± 5.91	
	**University**	52	34.0 ± 3.56	
	**Total**	91	32.6 ± 5.10	
**Weeks of Pregnancy**	**Primary**	1	28.0	*p* > 0.05
	**Secondary**	38	31.0 ± 5.85	
	**University**	52	31.2 ± 5.56	
	**Total**	91	31.1 ± 5.63	

SD: Standard Deviation.

**Table 5 ijerph-19-07797-t005:** Multiple linear regression models.

	Dependent Variable: Infant Weight				
Independent Variables	Non-Standardized B Coefficients	Standardized B Coefficients	95% Confidence Interval		Statistical Significance
**Constant**	−2.018		−3.941	−0.095	***p* < 0.05**
**Oral Hygiene Index (IHOS)**	−0.169	−0.307	−0.261	−0.077	***p* < 0.05**
**Week of delivery**	0.138	0.474	0.089	0.187	***p* < 0.05**

**Table 6 ijerph-19-07797-t006:** Results of different variables according to smoking habit.

	Smoking Habit	Frequency	Mean ± SD	Statistical Significance
**Caries index (CAOD)**	**Yes**	5	8.60 ± 4.615	*p* > 0.05
	**No**	88	6.93 ± 3.526	
**Simplified Oral Hygiene Index (IHOS)**	**Yes**	5	1.28 ± 0.715	*p* > 0.05
	**No**	88	1.08 ± 0.727	
**Week of delivery**	**Yes**	5	38.6 ± 2.61	*p* > 0.05
	**No**	88	39.2 ± 1.30	
**Infant weight (mg)**	**Yes**	5	2798 ± 0.5	***p* < 0.05**
	**No**	88	3235 ± 0.3	
**Number of Children**	**Yes**	5	1.0 ± 0.0	*p* > 0.05
	No	88	1.5 ± 0.62	
**Age**	Yes	5	30.2 ± 2.38	*p* > 0.05
	No	88	32.4 ± 5.26	
**Weeks of Pregnancy**	Yes	5	32.4 ± 4.82	*p* > 0.05
	No	88	31.1 ± 5.41	
**Correct Answers**	Yes	5	10.0 ± 3.39	*p* > 0.05
	No	88	12.7 ± 3.49	
**Incorrect Answers**	Yes	5	10.8 ± 1.92	*p* > 0.05
	No	88	8.6 ± 3.41	

SD: Standard Deviation.

## Data Availability

The data presented in this study are available on request from the corresponding author.

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
