# Peer review of "Relationship between Oral Health Knowledge and Maternal Oral Health with Obstetric Risk and Breastfeeding"

_ijerph, 2022, doi:10.3390/ijerph19137797_

Round 1
Reviewer 1 Report
An interesting read. My main concern is that the intro/aim and methods describes a study concerning dental health/knowledge and infant BW and the literature supports an association for these factors but breastfeeding which is a secondary thread throughout the manuscript gets little attention and the scientific rationale for including it in the manuscript seems lacking. Hence the narrative across the manuscript is lacking cohesion and clarity. Some of the associations found such as BW and gest weeks are not novel even when including oral information as is the association of smoking and birthweight. I think the paper would be better if those covariates were given less attention and the oral health tools and scales applied given a little more discussion. The limitations should also include potential biases in the sample compared to the general population, including estimates of ethnicity for the women interviewed, and a little more on the sociodemographic characteristics of the participants.
Some minor formatting/paragraph structure areas to be addressed highlighted in the attached pdf.

Author Response
COMMENTS ON THE MANUSCRIPT
=====================
Reviewer 1
=====================
1.-Reviewer’s comment:
Some of the associations found such as BW and gest weeks are not novel even when
including oral information as is the association of smoking and birthweight. I think the paper would be better if those covariates were given less attention and the oral health tools and scales applied given a little more discussion.
Authors’ response:
Thank you very much for this suggestion.
The reviewer indicates that some of the associations found, such as body weight and weeks of gestation, are not novel even when oral information is included, as is the association between smoking and birth weight. We agree that this is something known in the international literature, however, it has not been carried out in our health area and there are no Spanish studies on it, hence the value of our study, which we hope will be the beginning of other similar studies on the subject.
The Ethics and Clinical Research Committee authorized us to carry out the study for certain objectives and to collect certain variables, therefore, we must carry out the study with these limitations imposed by the regulations on clinical studies.
The oral health tools and scales used are widely known and validated in the literature, a comprehensive review of them would be a worthwhile work. Within our line of research, we intend to conduct a systematic review with meta-analysis on oral health knowledge and an outcome or dependent variable that is yet to be defined.
1.-Reviewer’s comment:
The limitations should also include potential biases in the sample compared to the general population, including estimates of ethnicity for the women interviewed, and a little more on the sociodemographic characteristics of the participants.
Authors’ response: Thank you very much for this suggestion.
The race variable was not included in the study as there is no racial or cultural diversity in our area of work. It is a predominantly Spanish population, with low immigration rates. The percentage of immigration is 2-3%, being predominantly Romanian and Hispanic-American (Ecuador, Peru), so this does not represent a bias or limitation in the study.
Should we have considered that a variable may have affected the representativeness of the sample, it would have been collected and analyzed; and a stratified analysis would have been carried out accordingly.
The following changes have been highlighted in yellow in the article:
Line 72
These two paragraphs have been merged. The bibliography has been merged.
Several systematic reviews [24,25] associate periodontitis in pregnancy with the risk of preterm delivery or the birth of low-birth-weight babies. Periodontal infections constitute a portal for the spread of bacteria that can reach the fetoplacental unit and jeopardize its stability [26-28]. This type of disease constantly releases Gram-negative anaerobic bacteria, bacterial products, endotoxins, lipopolysaccharides, and inflammatory mediators (PGE-2 and TNF alpha) into the bloodstream that, once disseminated by the transplacental hematogenous route, could lead to intrauterine infection-inflammation [29]. Increased production of inflammatory cytokines and c-reactive protein may contribute to uterine contraction leading to miscarriage, premature delivery, or low birth weight [30].
Line 82
The following two paragraphs have been merged:
Most clinical studies indicate a positive correlation between periodontal disease and possible complications in pregnancy due to microbiological and immunological factors [31-37]. According to the 2005 Oral Health Survey in Spain [17], 25% of pregnant women present periodontitis, these data coincide with other reviews that suggest a prevalence of between 5 and 20% [18].
Line 95
We have changed the main objective:
The main aim of the study was to analyze whether there is an association between the level of knowledge on oral health, maternal oral health status and breastfeeding and the possible influence on gestational age and neonatal weight.
Line 99
We have added a phrase about racial and ethnicity
A descriptive cross-sectional study was conducted in the Talavera de la Reina Health Area (Toledo, Spain), at an oral health care office of the Talavera-Centro Health Center (Castilla- La Mancha Health Service, SESCAM), between October 2017 and October 2019. A consecutive sample of pregnant women who attended the birth preparation sessions given by the midwife of the Health Center was used for data collection. These women were referred to the Oral Health Unit of the center, where they were informed of the study. In our area, there are no racial or ethnic differences that might bias or influence the results of the study.
Line 188
We have changed FIV to IFV
Collinearity was analyzed with the tolerance and IVF statistics. For no multicollinearity, the tolerance must be high; a tolerance less than 0.10 diagnoses serious collinearity problems. Moreover, IVF is a reciprocal indicator of tolerance. The lower this value, multicollinearity must be reduced. It is estimated that an IVF value greater than 10 is diagnostic of serious collinearity problems. A confidence level of 5% was established. The SPSS statistical program for Windows (Statistical Package Social Sciences version 29.0) and Epidat 4.1 for Windows were used for data analysis.
Line 246
We have changed the column “Frequency”
Table 2. Results of the different variables according to breastfeeding habits.
|
Breastfeeding |
F Frequency |
Mean ± SD |
Statistical significance |
Caries Index (CAOD) score |
Yes |
82 |
7.18±3.615 |
p > 0.05
|
No |
15 |
6.93±3.515 |
||
Simplified Oral Hygiene Index (IHOS) score |
Yes |
82 |
1.04±0.739 |
p > 0.05 |
No |
15 |
1.17±0.637 |
||
Week of delivery |
Yes |
82 |
39.1±1.44 |
p > 0.05 |
No |
15 |
39.3±0.81 |
||
Infant's weight (milligrams) |
Yes |
82 |
3221±0.41 |
p > 0.05 |
No |
15 |
3220±0.34 |
||
Number of Children |
Yes |
82 |
1.5±0.63 |
p > 0.05 |
No |
15 |
1.4±0.63 |
||
Mother's age in years |
Yes |
82 |
33.0±4.80 |
p < 0.05* |
No |
15 |
29.6±6.42 |
||
Weeks of Pregnancy |
Yes |
82 |
31.0±5.35 |
p > 0.05 |
No |
15 |
30.8±6.38 |
||
Correct answers "Questionnaire for pregnant women. Oral health" |
Yes |
82 |
12.8±3.40 |
p < 0.05* |
No |
15 |
10.7±4.02 |
||
Incorrect answers "Questionnaire for pregnant women. Oral health” |
Yes |
82 |
8.5±3.33 |
p > 0.05 |
No |
15 |
10.2±3.26 |
SD: Standard Deviation
Line 163
In section 2.2. Statistical analysis, information has been added for the ANOVA analysis, stating that it has been performed with the Bonferroni test of adjustment.
For the descriptive analysis, descriptive parameters were used according to the scale of the variable; qualitative variables were calculated using means of frequencies and percentages, and quantitative variables were resolved using central position statistics (mean) and standard deviation. The distribution analysis was analyzed using the Kolmogorov-Smirnov and Levene tests for the study of the normal distribution of the variable with the Lilliefors significance level, in addition to verifying the symmetry of the curve or observing the frequency histograms with the normal curve. For the inferential analysis, in the case of the analysis of independent variables, ANOVA was used to study the relationship between a normal continuous variable and a nominal variable or the study of n independent groups with the correction of Bonferroni; if the outcome variable was a dichotomous variable, the Student's t test was used. To compare nominal and dichotomous variables, the Chi-square test was used. Multiple linear regression models were constructed following the backward procedure, using infant weight as the dependent variable and the rest of the variables studied as independent variables. Multiple linear regression models were constructed following the backward procedure, using infant weight as the dependent variable and the rest of the variables studied as independent variables. The backward procedure was performed whit with the selection of a significance level to stay in the model (p < 0.05), fitting the model with all possible predictors and comparing these after removing the predictor.
We strongly believe that the reviewers’ recommendations have improved the quality of the manuscript and enhanced clarity and presentation.
Thank you for your recommendations.
Sincerely,
The Authors

Reviewer 2 Report
Dear Authors,
The article 'Relationship between oral health knowledge and maternal oral health with obstetric risk and breastfeeding' was to analyze the association between maternal oral knowledge and level of oral health during pregnancy with the risk of obstetric complications and breastfeeding. A descriptive cross-sectional study was conducted after an oral health educational intervention in a consecutive sample of 97 pregnant women.
Merge affiliations.
Minor spell check required. American English is required.
Punctuation mistakes should be corrected.
The introduction provides suffcient back ground and includes all relevant references.
Combine citations - Most clinical studies indicate a positive correlation between periodontal disease and possible complications in pregnancy due to microbiological and immunological factors [31,32,33,34,35,36,37]. SHOULD BE
Most clinical studies indicate a positive correlation between periodontal disease and possible complications in pregnancy due to microbiological and immunological factors [31-37].
Justify the text
p value should be written in italics
results
Prepare table using MDPI guidelines
Discussion is crealy presented.
Add a table with abbreviations before references.
Prepare references using MDPI guidelines
The article is well planned and prepared. It contains a decent summary of the analyzed topic.
Article can be accepted after major revisions.
Author Response
====================
Reviewer 2
=====================
2.-Reviewer’s comment; Merge affiliations.
Authors’ response:
Thank you very much for this suggestion. Unfortunately, we have not been able to reorganize and merge the authors' affiliations, because the authors' rules state that the authors' initials, ORCIDs and contact emails must appear, which is why there are different lines.
2.-Reviewer’s comment; Minor spell check required. American English is required.
Authors’ response:
Thank you very much for this suggestion. The entire text has been proofread by a native English speaker specialized in scientific writing. Edits have been made throughout the text to improve the English. See proofreading certificate attached.
2.-Reviewer’s comment; Punctuation mistakes should be corrected.
Authors’ response:
Thank you very much for this suggestion. As commented previously, the entire text has been revised and punctuation errors have been corrected. For example, in line 103, there is a comma instead of a full stop:
“these women were referred to the Oral Health Unit of the center, where they were in-formed of the study,”
2.-Reviewer’s comment; The introduction provides sufficient background and includes all relevant references.
Authors’ response:
Thank you for your comment. We have added aspects on the effect of breastfeeding on the introduction, in addition to the conclusion, based on the recommendation of reviewer 1.
2.-Reviewer’s comment; Combine citations.
Authors’ response:
Thank you for your comment. The recommended change has been made.
2.-Reviewer’s comment; Justify the text.
Authors’ response:
Text has been justified throughout the document.
2.-Reviewer’s comment; p value should be written in italics.
Authors’ response:
Thank you very much for this comment. Changes in the spelling of the p value have been made in the text.
2.-Reviewer’s comment; Results. Prepare table using MDPI guidelines.
Authors’ response:
Thank you very much for this comment. We have revised table 2 to conform to MDPI standards.
2.-Reviewer’s comment; Add a table with abbreviations before references.
Authors’ response:
Thank you for your comment. A table of abbreviations used in the text has been added before the references.
Abbreviations |
Significance |
CAOD |
Caries Index |
CEIC |
Clinical Research Ethics Committee |
CI-S |
Simplified Calculus Index |
DI-S |
Simplified Debris Index |
DW |
Durbin-Watson |
ICU |
Intensive Care Unit |
IHOS |
Simplified Oral Hygiene Index |
IVF |
Inflation Variance Factor |
SESCAM |
Health Service of Castilla La Mancha |
WHO |
World Health Organization |
2.-Reviewer’s comment: Prepare references using MDPI guidelines.
Authors’ response:
Thank you for your comment. We have revised the references using the guidelines.
We strongly believe that the reviewers’ recommendations have improved the quality of the manuscript and enhanced clarity and presentation.
Thank you for your recommendations.
Sincerely,
The Authors

Reviewer 3 Report
The authors report the data from a descriptive cross-sectional study aimed at evaluating the oral health awareness of pregnant women in a consecutive sample at a unique oral health care office. Beyond the limitations intrinsic to this study design, there is one major (reporting?) flaw that needs to be addressed. Some observations are given below to improve the quality of the manuscript.
Introduction
- General concepts about the relationship between periodontitis and gestational complications are presented, though I would suggest the authors to make an effort to make the reading more flowable and to avoid redundant statements (i.e. lines 83-85).
Material and methods
- Line 101, was this period elicited?
- Line 141, please add details on the type of probe used
- Line 143-146, please clarify why the authors adopted this partial mouth examination method
- It is unclear when and how the study outcome (infant weight) was measured. Was it self-reported? Since the main outcomes are related to infant characteristics, I presume the cross-sectional examination was carried out after infant delivery, but this information is not explicated throughout the manuscript. This point represent a major reporting flaw which must be fixed.
- Level of adjustment for multivariate analysis is not reported.
- Table 4, I am concerned of the legitimacy of using ANOVA analysis with only one participant in the group 'primary'. A non parametric test would have been more appropriate.
- Figure 2, what is the independent variable chosen for this linear regression?
Discussion
- Line 266, please express as Vozza et al., the same at line 282 for Garcia-Martin and Castaldi at line 287
- Line 297, the present reviewer still find very difficult to understand how the authors assessed breastfeeding if the study visit was administered during pregnancy.
Conclusion
- The second paragraph of the conclusion is more suitable for a narrative review than for the present cross-sectional study. Please, try to stick more with the actual findings
Author Response
=====================
Reviewer 3
=====================
3.-Reviewer’s comment: The authors report the data from a descriptive cross-sectional study aimed at evaluating the oral health awareness of pregnant women in a consecutive sample at a unique oral health care office. Beyond the limitations intrinsic to this study design, there is one major (reporting?) flaw that needs to be addressed. Some observations are given below to improve the quality of the manuscript.
Authors’ response:
Thank you for your comments. We will try to resolve all doubts with your comments and those of the other reviewers, which we hope will improve and clarify the text.
3.-Reviewer’s comment Introduction. General concepts about the relationship between periodontitis and gestational complications are presented, though I would suggest the authors to make an effort to make the reading more flowable and to avoid redundant statements (i.e. lines 83-85).
Authors’ response:
Thank you for your comment. Following the suggestion by Reviewer 1, we have merged the two paragraphs, to make them easier to read, we have also had a native English speaker read through the entire text to improve the English. See proofreading certificate attached.
3.-Reviewer’s comment: Material and methods. Line 101, was this period elicited?
Authors’ response:
Thank you for your comment. The study period is as indicated in line 99-102. The women were recruited and immediately reported.
A descriptive cross-sectional study was conducted in the Talavera de la Reina Health Area (Toledo, Spain), at an oral health care office of the Talavera-Centro Health Center (Castilla- La Mancha Health Service, SESCAM), between October 2017 and October 2019.
3.-Reviewer’s comment: please add details on the type of probe used
Authors’ response:
Thank you for your comment. We have changed the original line 146-153:
It consists of two elements: a simplified debris index (DI-S) and a simplified calculus index (CI-S).
Which we have replaced with:
It consists of two elements: a simplified debris index (DI-S) which describes the extent of soft deposits and is one of the 2 components of the simplified oral hygiene index (IHOS). The DI-S is obtained per person by totaling the debris score per tooth surface and dividing the result by the number of surfaces examined. Moreover, the simplified calculus index (CI-S). The average individual or group DI-S and CI-S are combined to obtain IHOS.
3.-Reviewer’s comment: Line 143-146, please clarify why the authors adopted this partial mouth examination method.
Authors’ response:
Thank you for your comment. This index has been used because it is a tool recognized by the WHO, which makes it easy to examine and record as it has only one observer and many other variables to be recorded. We have explained this in lines 143-146.
The IHOS is proposed by the WHO to evaluate the state of oral hygiene, assessing the presence and quantity of plaque and dental calculus, classified from 0 to 3. It determines and evaluates the degree of oral hygiene in the population, both quantitatively and qualitatively [44].
3.-Reviewer’s comment:
It is unclear when and how the study outcome (infant weight) was measured. Was it self-reported? Since the main outcomes are related to infant characteristics, I presume the cross-sectional examination was carried out after infant delivery, but this information is not explicated throughout the manuscript. This point represent a major reporting flaw which must be fixed.
Authors’ response:
Thank you for your comment. We have now explained this in the text:
Newborns were subsequently examined by the midwife at the health unit fifteen days after birth. During this visit, a general examination of the newborn and the mother was performed, including the newborn's birth weight as reflected in the hospital med-ical record and the mother was given a survey about breastfeeding.
3.-Reviewer’s comment: Level of adjustment for multivariate analysis is not reported.
Authors’ response:
Thank you for your comment. This is now included in line 175:
Multiple linear regression models were constructed following the backward procedure, using infant weight as the dependent variable and the rest of the variables studied as independent variables. Multiple linear regression models were constructed following the backward procedure, using infant weight as the dependent variable and the rest of the variables studied as independent variables. The backward procedure was per-formed whit with the selection of a significance level to stay in the model (p < 0.05), fit-ting the model with all possible predictors and comparing these after removing the predictor.
3.-Reviewer’s comment:
Table 4, I am concerned of the legitimacy of using ANOVA analysis with only one participant in the group 'primary'. A non parametric test would have been more appropriate.
Authors’ response:
Thank you for your comment. We decided to use ANOVA analysis as the data from the different groups followed a normal distribution, according to the normality tests used, and the Bonferroni adjustment was used. This aspect has been added in section 2.2. on statistics, following the recommendations of Reviewer 1.
3.-Reviewer’s comment: Figure 2, what is the independent variable chosen for this linear regression?
Authors’ response:
Thank you for your comment. We have clarified this in the text as follows:
In the multiple linear regression model (Table 5 and Figure 2) the variables that were statistically significant (p < 0.05) and related to infant weight were IHOS (Coefficient: -0.169), indicating that there was an inverse relationship between IHOS and birth weight (p < 0.05). Meanwhile, week of delivery had a direct relationship with birth weight (Coefficient: 0.138) (p < 0.05). In Figure 2 the independent variable IHOS and birth weight are shown.
We do not have a technical solution to include it in the same figure, therefore we have included the explanation in the text.
3.-Reviewer’s comment: Line 266, please express as Vozza et al., the same at line 282 for Garcia-Martin and Castaldi at line 287.
Authors’ response:
Thank you for your comment. We have followed the reviewer’s suggestions.
3.-Reviewer’s comment: Line 297, the present reviewer still finds very difficult to understand how the authors assessed breastfeeding if the study visit was administered during pregnancy.
Authors’ response:
Thank you for your comment. We have added the following text to clarify this:
Newborns were subsequently examined by the midwife at the health unit fifteen days after birth. During this visit, a general examination of the newborn and the mother was performed, including the newborn's birth weight as reflected in the hospital medical record and the mother was given a survey about breastfeeding.
3.-Reviewer’s comment: Conclusion. The second paragraph of the conclusion is more suitable for a narrative review than for the present cross-sectional study. Please, try to stick more with the actual findings
Authors’ response:
Thank you for your comment. We have made changes to the conclusions following the recommendations of the different reviewers.
This second paragraph is a project to be implemented once the results are known and is a proposal for the future.
We strongly believe that the reviewers’ recommendations have improved the quality of the manuscript and enhanced clarity and presentation.
Thank you for your recommendations.
Sincerely,
The Authors

Reviewer 4 Report
The authors interviewed 97 pregnant women, but at no time did they mention when they interviewed them again. The most important variables of the association are related to their children: birth weight and breastfeeding. So how could these data be obtained if interviewed pregnant women?
The authors use two indices to establish oral health status but do not mention inter-observer calibration. The authors should note how many observers participated and how they were calibrated.
It is necessary to define obstetric risk.
The introduction is extensive and should be shortened. The authors refer to possible associations between periodontal disease, premature birth, and low birth weight. However, their research strategy was based on an oral hygiene index that shows the gingival status of the subject and not the periodontal status. It is suggested to include it in the discussion because they did not use a periodontal index.
The bibliography is extensive and includes doctoral theses, which, without detracting from their academic merits, we prefer to replace with articles published in indexed journals.
Author Response
===================
Reviewer 4
=====================
4.-Reviewer’s comment: : The authors interviewed 97 pregnant women, but at no time did they mention when they interviewed them again. The most important variables of the association are related to their children: birth weight and breastfeeding. So how could these data be obtained if interviewed pregnant women?
Authors’ response:
Thank you for your comment. These questions aspects have already been addressed based on comments made by Reviewer 3. Moreover, we have edited the following paragraph to clarify this in the main manuscript:
Newborns were subsequently examined by the midwife at the health unit fifteen days after birth. During this visit, a general examination of the newborn and the mother was performed, including the newborn's birth weight as reflected in the hospital medical record and the mother was given a survey about breastfeeding.
4.-Reviewer’s comment: The authors use two indices to establish oral health status but do not mention inter-observer calibration. The authors should note how many observers participated and how they were calibrated.
Authors’ response:
Thank you for your comment. We have added the following text:
There was only one observer in this study (S.S.S.) who is a qualified dentist with fifteen years of professional experience.
4.-Reviewer’s comment: It is necessary to define obstetric risk.
Authors’ response:
Thank you for your comment. We have added an explanation of obstetric risk:
Periodontal disease is currently considered an "emerging risk factor" for obstetric complications. These mainly include low birth weight (< 2500 g), preterm birth (< 37 weeks), miscarriage, stillbirth, and pre-eclampsia (commonly defined as maternal hypertension and proteinuria after the 20th week of gestation) [19]. However, there is no consensus on whether this association is consistent because previous studies show biases in patient characteristics and some divergence in the criteria used in the periodontal study [20].
4.-Reviewer’s comment: The introduction is extensive and should be shortened. The authors refer to possible associations between periodontal disease, premature birth, and low birth weight. However, their research strategy was based on an oral hygiene index that shows the gingival status of the subject and not the periodontal status. It is suggested to include it in the discussion because they did not use a periodontal index.
Authors’ response:
Thank you for your comment. Reviewer 1 has requested extensions to the introduction, while reviewer 2 felt that it was sufficient. Finally, the authors have decided to follow the comments of reviewer 1, so that the text does not lose information.
4.-Reviewer’s comment:
“It is suggested to include it in the discussion because they did not use a periodontal index.”
Authors’ response:
Thank you for your comment. We have added the following text in the discussion:
The IHOS index was used because it is recognized by the WHO for this type of study. Furthermore, it is noninvasive and easy to explore and record since it only requires one observer. It would be interesting to use other periodontal indexes in future studies.
4.-Reviewer’s comment: The bibliography is extensive and includes doctoral theses, which, without detracting from their academic merits, we prefer to replace with articles published in indexed journals.
Authors’ response:
Thank you for your comment. We have changed the reference:
- Rodriguez, M. Nivel de conocimiento sobre prevención en salud bucal en gestantes del Hospital Nacional Daniel A. Carrión. Tesis. Universidad Nacional Mayor de San Marcos. 2002. Perú.
Which we have replaced with the following:
- Penmetsa G.S.; Meghana K.; Bhavana P.; Venkatalakshmi M.; Bypalli V.; Lakshmi B.. Awareness, attitude and knowledge regarding oral health among pregnant women: A comparative study. Niger Med J 2018;59:70-3. https://doi.org/10.4103/nmj.NMJ_151_18
We strongly believe that the reviewers’ recommendations have improved the quality of the manuscript and enhanced clarity and presentation.
Thank you for your recommendations.
Sincerely,
The Authors

Round 2
Reviewer 2 Report
Some editorial corrections are needed.
Article can be accepted after Editor decision.
Reviewer 3 Report
The authors have addressed all major requests. Except for the table layout, I would recommend this paper suitable for publication.
Reviewer 4 Report
The authors attended to the comments and suggestions requested.